# Understanding the Pathophysiology of Ischemic Stroke: The Basis of Current Therapies and Opportunity for New Ones

**DOI:** 10.3390/biom14030305

**Published:** 2024-03-04

**Authors:** Maryam A. Salaudeen, Nura Bello, Rabiu N. Danraka, Maryam L. Ammani

**Affiliations:** 1Division of Neuroscience, Faculty of Biology, Medicine and Health, School of Biological Sciences, University of Manchester, Manchester M13 9PL, UK; 2Department of Pharmacology and Therapeutics, Faculty of Pharmaceutical Sciences, Ahmadu Bello University, Zaria 810107, Nigeria; bnura@abu.edu.ng (N.B.); rndanraka@abu.edu.ng (R.N.D.); 3Department of Pharmacology and Therapeutics, Faculty of Clinical Sciences, College of Health Sciences, Usmanu Danfodiyo University, Sokoto 840001, Nigeria; 4Department of Biosciences, School of Science and Technology, Nottingham Trent University, Nottingham NG11 8NS, UK; maryam.ammani@kasu.edu.ng; 5Department of Pharmacology and Toxicology, Faculty of Pharmaceutical Sciences, Kaduna State University, Kaduna 800283, Nigeria

**Keywords:** ischemic stroke, neuroinflammation, oxidative stress, neuroprotection, cellular therapy, drug repurposing

## Abstract

The majority of approved therapies for many diseases are developed to target their underlying pathophysiology. Understanding disease pathophysiology has thus proven vital to the successful development of clinically useful medications. Stroke is generally accepted as the leading cause of adult disability globally and ischemic stroke accounts for the most common form of the two main stroke types. Despite its health and socioeconomic burden, there is still minimal availability of effective pharmacological therapies for its treatment. In this review, we take an in-depth look at the etiology and pathophysiology of ischemic stroke, including molecular and cellular changes. This is followed by a highlight of drugs, cellular therapies, and complementary medicines that are approved or undergoing clinical trials for the treatment and management of ischemic stroke. We also identify unexplored potential targets in stroke pathogenesis that can be exploited to increase the pool of effective anti-stroke and neuroprotective agents through de novo drug development and drug repurposing.

## 1. Introduction

For over three decades, researchers have had tremendous success unravelling the molecular and cellular changes that occur following a stroke. This breakthrough also accurately identifies the various risk factors for stroke. Typically, stroke is a neurological disorder that results from a partial or complete shortage of blood supply to any part of the brain. The shortage of blood supply is often the consequence of obstructed blood flow (ischemic stroke, IS) or blood leakage from a ruptured cerebral blood vessel (haemorrhagic stroke). In the face of limited or absent blood supply, the affected brain region(s) suffer oxygen and nutrient deprivation and become necrotic. In the absence of prompt intervention, this event is followed by a cascade of events that can cause the death of surrounding tissues, known as the ischemic penumbra. Stroke treatment thus occurs via three major approaches—preventive measures, management measures, and post-stroke rehabilitation measures [1]. The only FDA- and EU-approved drug for IS treatment, alteplase, has a short time window of 3–4.5 h and leaves a post-treatment side effect of intracerebral haemorrhage [2,3,4]. Since its discovery over two decades ago, alteplase remains the most effective thrombolytic agent for acute IS. There is therefore an urgent need to explore potential therapies. A clear understanding of the causes and pathophysiology of ischemic stroke presents a unique opportunity for the development of a pool of potential therapies for the condition. In this review, we dive deep into ischemic stroke pathophysiology and identify possible targets for novel therapies including cellular therapies. We also make recommendations for drug repurposing and redesign of some existing drugs with potential for use in IS.

## 2. Etiology and Pathophysiology of Ischemic Stroke

The causes (etiology), pathogenesis, and the underlying molecular, biochemical, and structural changes that occur in the face of any disease condition constitute its pathophysiology. Ischemic stroke occurs almost suddenly within minutes of blood supply interruption to brain tissues due to blockage of arteries supplying the brain by either blood clots formed by atrial fibrillation or thrombus formed on fatty deposits referred to as atherosclerotic plaque [5]. The affected brain region is often regarded as the ischemic core. Here, most of the cells undergo irreversible death before the effect(s) of neuroprotective agents are established. Surrounding the ischemic core is a region of salvageable cells known as the ischemic penumbra that often constitutes the target of therapeutic interventions. An interplay between complex molecular and cellular mechanisms results in some phenotypic manifestations including hemiplegia, paraplegia, dysarthria, and paresis. Other manifestations may occur depending on the region of the brain that receives blood supply from the occluded arteries [6]. Similar to numerous other neurodegenerative conditions, ischemic stroke is characterized by a multitude of changes within the afflicted ischemic core and the surrounding penumbra. These macro- and microscopic changes are commonly categorized under five overarching terms: Neuroinflammation, Excitotoxicity, Oxidative stress, Apoptosis, and Autophagy (Figure 1). Cell death in ischemic stroke occurs due to complex interactions between these independent but mutually reinforcing series of pathological events. 

### 2.1. Excitotoxicity

Continuous blood supply is critical to the survival of the brain because the brain constantly requires oxygen and nutrients for proper functioning, relying on effective blood circulation [7]. Once a major cerebral artery is blocked, the blood supply to the affected brain region is reduced. The diminished circulation causes energy disruption due to hypoxia and ischemia by interfering with ATP production. Consequently, ionic gradients of ion channels, including calcium ATPase, sodium/calcium exchange, and sodium/potassium ATPase on plasma and organelle membranes of neurons are disrupted [8]. This leads to an excess influx of calcium into the neurons, and the activation of calcium ion-dependent enzymes ultimately causes the release of excess glutamate and a reduction in its reuptake [9,10]. This series of events constitutes excitotoxicity due to excessive stimulation of N-methyl-D-Aspartate receptors (NMDAR) in the membrane of postsynaptic neurons leading to the generation of reactive oxygen species (ROS) causing oxidative stress, which then interrupts mitochondrial function and neuronal death occurs [11]. Excessive activation of NMDA receptors also contributes to the disruption of neuronal plasticity, affecting aging, memory, and learning, which leads to cognitive decline associated with stroke [12].

### 2.2. Oxidative Stress

When blood flow to the brain is disrupted, it leads to impaired energy metabolism and oxidative stress injury. Recanalization and reperfusion following blood flow obstruction have been established as culprits in oxidative stress-induced injury. Oxidative stress, a major mechanism in ischemic stroke, disrupts the oxidant–antioxidant balance, particularly in brain cells rich in polyunsaturated fatty acids. Factors such as low antioxidants, high pro-oxidants (e.g., iron), and elevated oxidative metabolism contribute to worsened oxidative damage [13].

Acute ischemic stroke disrupts calcium homeostasis, releasing calcium in the brain and activating pathways that produce ROS and oxidative damage. This imbalance in oxidants and antioxidants results in excessive ROS and hydroxyl radicals, causing extensive damage to the brain [14]. Cellular ROS generation further increases during ischemic stroke due to glucose and oxygen deprivation, exacerbating oxidative stress and brain damage [15]. Superoxide anion production during ischemia is attributed primarily to xanthine oxidase (XO) and NADPH oxidase (NOX). ATP depletion during ischemia causes an accumulation of hypoxanthine and xanthine, substrates for XO, leading to ROS generation [16,17]. Increased XO expression in the infarcted area after an ischemic stroke has been observed. NOX, another significant source of ROS, is upregulated post-stroke, with NOX2 identified as the primary source of superoxide production activated by the N-methyl-D-aspartate receptor [18,19].

Mitochondria recognized as the cellular powerhouses, play a crucial role in maintaining cell energy homeostasis, making them integral players in ischemic neuronal death. The breakdown of mitochondrial respiratory function and membrane potential triggers a cascade of events leading to neuronal demise after ischemia. The depolarization of mitochondria initiates excessive production of ROS, decreased ATP generation, and the accumulation of PTEN-induced putative kinase 1 (*PINK1*) and unfolded protein response (*UPR*) [20]. As ROS levels rise and calcium overloads, the membrane permeability transition pore (MPTP) opens, releasing cytochrome c. This activation triggers effector caspases, ultimately executing apoptotic death [21]. *PINK1*, in response to mitochondrial damage, recruits Parkin and phosphorylates both Parkin and ubiquitin, initiating mitophagy [22].

### 2.3. Neuroinflammation

Neuroinflammation involving several immune cells, such as innate immune cells and adaptive immune cells also plays a crucial role in IS. The brain insult that follows ischemic stroke results in necrosis and apoptosis, driving an inflammatory reaction controlled by the discharge of ROS, chemokines, and cytokines. This process springs up in the microcirculation and involves several cytotypes, such as innate immune cells (i.e., the microglia) and adaptive immune cells (i.e., lymphocytes), causing neuronal death [23]. The neuroinflammation process depends on the scene, period, and course of the neurological insult. Microglia play a dual role in neuroinflammation during the acute phase of stroke onset. MicroRNAs, such as miR-203, have been found to mitigate cerebral ischemia–reperfusion injury by targeting microglia [24]. Furthermore, the polarization of microglia, particularly M1 polarization, has been associated with exacerbating cerebral ischemia [25]. The intense neuroinflammation during the acute phase of stroke is linked to blood–brain barrier (BBB) breakdown, neuronal injury, and poor outcomes [26]. Neuronal death is the ultimate determinant of IS-induced morbidity and mortality, and the success of its management is determined by the extent to which it is prevented.

### 2.4. Apoptosis

Apoptosis involves a series of intrinsic and/or extrinsic events that shrink the neurons and condense the cytoplasm ultimately breaking their nuclear membrane to form apoptotic bodies. In the intrinsic pathway, reduced nutrients and oxygen supply to the cell disrupts ATP production by the normal glycolytic oxidative phosphorylation pathway. As such, the anaerobic pathway predominates, and the ATP produced is insufficient to maintain cellular activities. This results in ionic imbalance (Na^+^/Ca^2+^ influx and K^+^ efflux) and calcium ion accumulates in the cell, which causes excessive release of excitatory amino acid neurotransmitters, especially glutamate, into the extracellular space. This process is then followed by a cascade of cytotoxic events in the nucleus and cytoplasm, including activation of calpain (calpain-mediated), generation of ROS (reactive oxygen species-mediated) from mitochondrial metabolism, which causes damage to the cellular membrane, and DNA breakage (DNA damage-mediated events) [27], while the extrinsic pathway, which often occurs independently or in synergy with the intrinsic pathway, involves the activity of the signaling factors of inflammation released by astrocytes, microglia, and oligodendrocytes due to cerebrovascular damage. These inflammatory signaling factors include various proinflammatory cytokines and receptors including TNF-α/β, chemokines, interleukin 1β, TNF-related apoptosis-inducing ligand receptor (TRAIL-R), and Fas ligand (*FasL*) [28]. These receptors at the neuronal cell membrane trigger an apoptotic event involving a signal induced by caspase-8 activating the downstream effector caspase-3 or BID, which mediates apoptosis through the mitochondrial-dependent pathway [29].

In addition to apoptosis, cell death following an ischemic stroke can also occur via any of these five mechanisms: ferroptosis, phagoptosis, parthanatos, pyroptosis, and necroptosis (Figure 2). Understanding the intricate interplay of these different cell death pathways in the context of ischemic stroke is crucial for developing targeted therapeutic interventions. Combining insights from these mechanisms could pave the way for more effective strategies to mitigate neuronal damage and improve outcomes in ischemic stroke patients.

#### 2.4.1. Ferroptosis

A recently defined form of cell death that has been implicated in the pathogenesis of ischemic stroke is ferroptosis. It is characterized by the accumulation of lipid peroxides and iron-dependent ROS, leading to oxidative damage and cell death [30]. In IS, ferroptosis has been shown to contribute to neuronal death and tissue damage [31]. For example, the level of soluble tau protein, which mediates iron transport, decreases in the ischemic region after stroke, leading to iron accumulation and neuronal death [32]. Inhibition of ferroptosis using specific inhibitors has been shown to protect against neuronal damage in animal models of stroke [33].

The regulation of iron metabolism and lipid peroxidation are key factors in the development of ferroptosis. Excessive iron accumulation and impaired antioxidant defense mechanisms, such as reduced glutathione peroxidase 4 (GPx4) activity, can promote lipid peroxidation and trigger ferroptotic cell death. Therefore, targeting iron metabolism and lipid peroxidation pathways may represent potential therapeutic strategies for ischemic stroke [34].

#### 2.4.2. Necroptosis

Necroptosis is a regulated form of necrosis that occurs in response to various stimuli, including ischemia and inflammation. It is mediated by the activation of receptor-interacting protein kinase 1 (*RIPK1*) and *RIPK3*, which ultimately leads to the phosphorylation and activation of mixed lineage kinase domain-like protein (MLKL). MLKLs then translocate to the plasma membrane, disrupting membrane integrity and cell death [35]. Necroptosis has been shown to contribute to neuronal death in ischemic stroke [36,37,38,39,40]. Inhibition of necroptosis through RIPK1 pharmacologic and genetic inhibition has been found to reduce neuronal damage and improve functional outcomes in animal models of ischemic stroke [41]. However, the exact role of necroptosis in ischemic stroke is still not fully understood, and further research is needed to elucidate its precise contribution to IS pathogenesis. Clinical evidence regarding the efficacy of necroptosis inhibitors for ischemic stroke is also limited

#### 2.4.3. Pyroptosis

This is a form of programmed cell death that is implicated in neuronal death during IS. Pyroptosis is mediated by the activation of caspase-1, triggered by the formation of inflammasomes in response to cerebral ischemia [42]. Inflammasomes are multi-protein complexes that consist of sensor proteins NLRP1, NLRP3, and NLRP4 that play a role in processing pro-inflammatory cytokines [43]. During cerebral ischemia, the activation of inflammasomes leads to the activation of caspase-1, which cleaves pro-IL-1β and IL-17 to produce IL-1β and IL-17, respectively, both of which are key inflammatory cytokines, inducing neuronal death along with other pro-inflammatory factors during pyroptosis [44]. Therefore, the mechanisms by which pyroptosis contributes to neuronal death in IS is believed to be the release of pro-inflammatory factors and the activation of inflammatory pathways, leading to neuroinflammation and exacerbating the damage caused by ischemia [45].

#### 2.4.4. Parthanatos

This is a form of regulated cell death that may be a possible mechanism of neuronal death in IS [46]. It is dependent on the PARP1 enzyme (poly ADP-ribose polymerase 1) and is activated by oxidative stress-induced DNA damage and chromatinolysis. Unlike apoptosis, parthanatos does not result in the formation of apoptotic bodies and small DNA fragments, but occurs without cell swelling and is accompanied by plasma membrane rupture [47]. PARP1, which is a nuclear, chromatin-associated protein, plays a critical role in parthanatos by recognizing and repairing DNA breaks through the poly (ADP-ribosyl)ation process that utilizes nicotinamide adenine dinucleotide (NAD+) and ATP [48]. Parthanatos is characterized by the depletion of NAD+ and the inhibition of glycolytic enzyme hexokinase, leading to necrosis, while excessive PARP1 activity and NAD+ depletion further impair cellular metabolic processes, promoting cell death [49].

#### 2.4.5. Phagoptosis

The mechanism of phagoptosis in IS involves the recognition, engulfment, and digestion of neurons by microglia. Microglia constantly monitor the surface of neurons and can recognize and engulf neurons that expose “eat me” signals. One of the typical “eat me” signals is the presence of phosphatidylserine (PS) on the cell surface [50]. PS expression may occur due to oxidative stress, increased calcium levels, or ATP depletion [51]. Neurons expressing PS are recognized by opsonins such as milk fat globule EGF-like factor 8 (MFG-E8) and vitronectin receptors or growth arrest-specific factor 6 and Mer receptor tyrosine kinase (MERTK) receptors [52,53]. Additionally, the presence of calreticulin on the cell surface, as a result of ER stress, can serve as an “eat me” signal [54]. Microglia and astrocytes produce C1q and C3 to induce phagocytosis in neuronal cells [55,56].

Cell death following ischemic stroke can occur via any of the five mechanisms: ferroptosis, phagoptosis, parthanatos, pyroptosis, and necroptosis. Understanding the intricate interplay of these different cell death pathways in the context of ischemic stroke is crucial for developing targeted therapeutic interventions. Combining insights from these mechanisms could pave the way for more effective strategies to mitigate neuronal damage and improve outcomes in ischemic stroke patients. Ferroptosis is a form of regulated cell death characterized by the iron-dependent accumulation of lipid peroxides, leading to membrane damage and cell demise. The disruption of cellular homeostasis during the ischemic event can induce the accumulation of ROS and iron, often exacerbated by ischemia–reperfusion injury, triggering lipid peroxidation. This, in turn, damages cell membranes and contributes to neuronal death. In phagoptosis, cells are eliminated through phagocytosis without undergoing the typical morphological changes associated with apoptosis. IS can induce phagoptosis as part of the neuroinflammatory response where activated microglia and macrophages phagocytose stressed or damaged neurons, thus clearing cellular debris. In parthanatos, cell death is triggered by the overactivation of PARP and subsequent energy depletion. Oxidative stress and DNA damage can lead to PARP activation, resulting in the excessive consumption of NAD+ and ATP, leading to energy failure and cell death. Pyroptosis is slightly similar to phagoptosis in terms of the involvement of inflammation. Pyroptosis is a highly inflammatory form of programmed cell death involving the release of pro-inflammatory cytokines and cell swelling due to the activation of inflammasomes. The release of pro-inflammatory cytokines such as interleukin-1β amplifies neuroinflammation, exacerbating ischemic injury. Last is necroptosis, a regulated form of necrosis involving receptor-interacting protein kinases (RIPK) and a mixed-lineage kinase domain-like protein (MLKL). IS- can induce necroptosis through various signaling pathways. Activation of death receptors and subsequent RIPK activation leads to MLKL phosphorylation, causing plasma membrane rupture and cell death. Key: 4-HNE = 4-Hyfroxy-2-nonenal, ADP = Adenosine diphosphate, AIF = Apoptosis-inducing factor, PARP = Poly(ADP-ribose) polymerase, ATP = Adenosine triphosphate, DAMPs = Damage-associated molecular patterns, GPX4 = Glutathione peroxidase 4, GSDM-D = Gasdermin D, IL = Interleukin, IS = Ischemic stroke, MDA = Malondialdehyde, MLKL = Mixed-lineage kinase domain-like protein, MLKLp = Phosphorylated MLKL, NLRP = NOD-like receptor pyrin domain-containing, PS = Phosphatidylserine, RIPK = Receptor-interacting protein kinases, ROS = Reactive oxygen species, TNFR1 = Tumour necrosis factor receptor 1. Arrows with flat ends = Inhibition.

### 2.5. Autophagy

Autophagy is a cellular process involved in the degradation and recycling of damaged or unnecessary cellular components through packaging into auto-phagosomes and assembling in the lysosomes, which plays a significant role in ischemic stroke [57]. Autophagy forms part of the cellular cascade of events triggered by oxygen and nutrient deprivation in IS [58]. During cerebral ischemia, the limited availability of insulin and amino acids prevents the activation of the mammalian target of rapamycin complex 1 (*mTORC1*), a primary inhibitor of autophagy [59]. This, along with the increased ratio of adenosine monophosphate (AMP) to adenosine triphosphate (ATP), activates AMP-activated protein kinase (AMPK), which enhances autophagy [60]. Additionally, mitochondrial dysfunction, accumulation of reactive oxygen species (ROS), and endoplasmic reticulum (ER) stress also induce autophagy in response to cerebral ischemia [61,62]. Hypoxia-inducible factor 1α (HIF-1α) signaling pathway is another mechanism through which autophagy is induced in ischemic stroke. *HIF-1α*, activated during hypoxia, can induce the transcription of genes involved in autophagy. It responds to systemic oxygen levels and promotes the transcription of genes that increase oxygen delivery to hypoxic regions [63,64]. Autophagy has been shown to have both beneficial and detrimental effects in ischemic stroke. On the one hand, autophagy can help remove damaged cellular components and promote cell survival. On the other hand, excessive or prolonged autophagy can lead to excessive degradation of cellular components and contribute to neuronal death [65]. 

## 3. Current Therapies for Ischemic Stroke and Their Targets

### 3.1. Thrombolytics/Thrombolytic Agents

Acute ischemic stroke is primarily treated via intravenous thrombolysis and sometimes followed by endovascular thrombolysis to enhance vessel recanalization [66]. The intravenous thrombolytic (IVT) treatment paradigm was originally developed to treat coronary thrombolysis but was later found to be effective in treating stroke patients. The efficacy of thrombolytic drugs depends on many factors including the age of the clot, the specificity of the thrombolytic agent for fibrin, and the presence and half-life of neutralizing antibodies [67]. Intravenous thrombolysis is achieved using alteplase, a second-generation thrombolytic, for the dissolution of blood clots. Other thrombolytics that are undergoing clinical trials and have shown comparable safety and efficacy to alteplase are prourokinase [68], tenecteplase [69], and staphylokinase [70]. Except for prourokinase which is an intra-arterial thrombolytic, these fibrinolytics are administered intravenously and represent a potential future alternative to alteplase. Endovascular thrombolysis is utilized for unclogging large vessels, and it is sometimes used as an add-on to intravenous thrombolysis [71,72]. Typically, these thrombolytics are plasminogen activators that aim to promote fibrinolysin formation. They act by converting plasminogen to soluble plasmin, a proteolytic enzyme that breaks down fibrin and fibrinogen in the thrombi blocking the affected cerebral vessel(s) [73].

In addition to endovascular thrombolysis, mechanical thrombectomy (MT) is becoming routine in many countries for large vessel occlusion (LVO) stroke [74]. Despite high rates of successful recanalization (≈85%) however, about 50% of patients do not reach functional independence at 3 months [75]. Adjunctive anti-thrombotic therapy might improve angiographic reperfusion by reducing the risk of distal emboli and arterial re-occlusion but is likely to expose patients to a higher intracranial hemorrhage (ICH) risk. Moreover, the concept of incomplete microvascular reperfusion (IMR), derived from observations of focal no-reflow following focal ischemia, may partially explain poor outcomes even after fast and complete proximal reperfusion [76]. Experimental utilization of antithrombotic agents has shown a reduction in IMR and improved outcomes [77].

### 3.2. Adjunctive Therapies

The reperfusion of the affected brain region(s) following an IS attack often occurs later than the optimal time needed to prevent damage to surrounding tissues. Complete recanalization is further made practically impossible by the presence of thrombi in smaller and non-visualized cerebral vessels consequently causing the formation of ischemic penumbrae. Many therapeutic approaches—antithrombotic agents, cellular therapy, and cytoprotectants—exist to salvage the ischemic penumbra. These strategies improve the speed and extent of reperfusion, promote neural repair and remodeling, and prevent or delay the deterioration of ischemic penumbra.

#### 3.2.1. Anti-Thrombotic Agents

In addition to the use of thrombolytics, other drugs known as anti-thrombotic agents are given to IS patients to prevent new clots from forming, thus enhancing the efficacy of alteplase and any other administered thrombolytics. These drugs ensure complete vessel recanalization by preventing clot formation via glycoprotein IIb/IIIa inhibition. Argatroban, glenzocimab, tirofiban, and eptifibatide are anti-thrombotic agents at different clinical trial phases used as adjuncts to intravenous thrombolysis with alteplase [78,79,80,81]. 

#### 3.2.2. Antiplatelet Therapy

This therapy is used for acute IS management and prevention of stroke incidence. It is also vital in controlling non-cardioembolic IS and transient ischemic attack (TIA). Antiplatelet agents like aspirin, clopidogrel, and ticagrelor are the most widely administered drugs to stroke sufferers within the first few days of attack [82]. They play a crucial role in the management of ischemic stroke by inhibiting platelet aggregation and preventing the formation of blood clots. These drugs act primarily by interfering with platelet activation and the coagulation cascade. Cyclooxygenase (COX) is responsible for converting arachidonic acid into prostaglandins, which play a role in platelet activation and aggregation. Aspirin irreversibly inhibits platelet COX, thereby decreasing the synthesis of thromboxane A2, a potent platelet aggregator. Clopidogrel is a thienopyridine derivative that irreversibly inhibits the P2Y12 adenosine diphosphate (ADP) receptor on the platelet surface, thus inhibiting ADP-induced platelet aggregation. Ticagrelor acts in a similar fashion to clopidogrel, but in a reversible manner. 

Dual antiplatelet therapy, which involves the combination of clopidogrel, prasugrel, or ticagrelor with aspirin, has become popular; many studies have tested the efficacy and safety of this dual therapy. Trial outcomes have suggested that clopidogrel and aspirin combination therapy is most beneficial if introduced within 24 h of stroke and continued for 4–12 weeks [83]. Multiple studies on antiplatelet therapy revealed that clopidogrel, alone or combined with aspirin, proves more effective in preventing blood clot formation during acute coronary syndrome and after percutaneous coronary intervention compared to aspirin alone [84,85,86]. On the contrary, the MATCH trial concluded that combining aspirin with clopidogrel did not reduce the risk of secondary stroke compared to using clopidogrel alone. Instead, the combination increased the likelihood of life-threatening bleeding complications. As a result, the trial did not recommend using clopidogrel and aspirin together for preventing recurrent ischemic stroke after an initial event or TIA [86,87]. These findings were consistent with the CHARISMA trial findings that compared the efficacy and safety of the combination of low-dose aspirin (75–162 mg/day) and clopidogrel (75 mg/day) versus low-dose aspirin alone in patients with high risk of atherothrombotic cerebrovascular events and prior documented vascular disease, including MI, stroke, TIA, and symptomatic peripheral vascular disease [88,89].

#### 3.2.3. Fibrinogen-Depleting Agents

High fibrinogen levels in stroke patients have been consistently linked to poor prognosis and unfavorable clinical outcomes, as indicated by research findings [90,91,92,93,94]. Fibrinogen-depleting agents decrease blood plasma levels of fibrinogen, hence reducing blood thickness and increasing blood flow. They also remove the blood clots in the artery and restore blood flow in the affected regions of the brain. Although some randomized clinical trials of defibrinogenation therapy identified beneficial effects of fibrinogen-depleting agents in stroke patients, others failed to show positive effects on clinical outcomes following stroke [95]. Moreover, some studies reported bleeding after treatment with defibrinogenating agents. Ancrod is a defibrinogenating agent derived from snake venom that has been studied for its ability to treat IS within three hours of onset [91]. The European Stroke Treatment with Ancrod Trial (ESTAT) concluded that controlled administration of ancrod at 70 mg/dL fibrinogen was efficacious, safe, and achieved a lower prevalence of ICH than observed at lower fibrinogen levels [92,96]. 

### 3.3. Cellular Therapies for Ischemic Stroke: A Paradigm Approach

Cellular therapies are important therapeutic options to be considered in the management of ischemic stroke due to their overwhelming effect in improving patient recovery. Stem cells can restore damaged brain tissue and neuronal cell loss, as well as reduce neuroinflammation. Mesenchymal stem cells, hematopoietic stem cells (blood stem cells), neural stem cells, and epithelial stem cells constitute the four main cells that have been tried in stem cell therapy (SCT) for the treatment of IS. The popularity and general acceptance of SCTs stems from their unique advantage of promoting tissue repair and regeneration, owing to their ability for self-renewal and multilineage differentiation [6]. SCTs offer promising therapeutic opportunities, safety, and efficacy to stroke patients. Research on embryonic stem cells, mesenchymal cells, and induced pluripotent stem cells has assessed their potential for tissue regeneration, maintenance, migration, and proliferation, rewiring of neural circuitry, and physical and behavioral rejuvenation [97]. Recently, a new type of mesenchymal stem cells (MSCs), called multilineage differentiating stress-enduring (Muse) cells, has been found in connective tissues. These cells offer great regenerative capacity and have been tested as a stroke treatment. After intravenous transplantation of Muse cells in a mouse model, they were found to engraft into the damaged host tissue and differentiate to provide functional recovery in the host [98]. Careful experimental design and clinical trials of stem cell therapies are likely to usher in a new era of treatment for stroke by promoting neurogenesis, rebuilding neural networks, and boosting axonal growth and synaptogenesis [1,99]. Additionally, amelioration and inhibition of one or more of the earlier mentioned overarching components of IS constitute the mechanism of neuroprotection and neurorepair properties of SCT [100,101,102] (Table 1).

Neural repair is an alternative therapy to neuroprotection. It is used to rejuvenate the tissue when the damage is already done and is therefore not time-bound but is most effective when administered 24 h after a stroke attack. Many animal models have been used in an attempt to stimulate neurogenesis and initiate the neuronal repair process [103]. Neural repair utilizes stem cell therapy to initiate repair mechanisms through cell integration into the wound or the use of neurotrophic factors to block neuronal growth inhibitors. These cells may be channeled to any injured region to facilitate greater synaptic connectivity. Clinical trials using neural stem cells have proven beneficial in stroke patients.

This table summarizes some clinical trials at different stages where stem cells are utilized for ischemic stroke. The trials are mainly focused on the safety and efficacy profile of these stem cells.

#### Limitations of Stem Cell Therapy and Way Forward

Stem cell therapy for IS, while promising, faces several limitations that currently hinder its widespread application. One significant challenge is the variable efficacy observed among individuals, with not all patients experiencing substantial improvements [104,105]. The factors influencing this variability remain poorly understood. Moreover, concerns about the safety of stem cell therapy persist, including the potential for teratoma formation and unintended tissue differentiation [106,107,108]. Ensuring the long-term safety of stem cell treatments is crucial for gaining regulatory approval and patient acceptance.

Determining the optimal timing, route, and dosage of stem cell administration poses another complex issue. The therapeutic window for effective treatment may be narrow, and the ideal dose and type of stem cells are still under investigation. There have been suggestions that lower cell doses (<10^7^) are preferred for use in chronic IS stages and vice versa [109]. Additionally, the question of immunorejection arises when stem cells from foreign donors are used, potentially requiring the use of immunosuppressive drugs and introducing additional risks and complications. This has led to debates in support of the use of close alternatives such as stem cell-derived conditioned medium, and extracellular vesicles such as exosomes and microRNAs [110,111,112].

Ethical considerations surround the use of embryonic stem cells, and obtaining sufficient quantities of adult stem cells for transplantation can also be challenging. While induced pluripotent stem cells (iPSCs) offer an alternative, concerns about their potential tumorigenicity persist. The delivery of stem cells to the damaged brain tissue is also a critical aspect that requires improvement. Ensuring proper migration, integration, and differentiation of transplanted stem cells into the affected areas remains a complex task. Scientists are exploring genetic modifications, preconditioning, and co-administration of supportive factors to improve the overall success of SCT.

## 4. Emerging Neuroprotective Agents for Ischemic Stroke: Pathophysiology-Targeted Therapies

The protection of ischemic penumbra from post-stroke degeneration and damage is ensured using neuroprotective and, more recently, neuroreparative agents. These agents act primarily by targeting one or several aspects of the underlying pathophysiology of IS thus increasing the likelihood of developing effective add-on neuroprotective therapies. In the subsequent paragraphs, we discuss neuroprotective drugs that target the various cascades in IS pathophysiology.

Prevention/reduction of Neuroinflammation: Preventing the activation of pro-inflammatory microglia phenotype holds promise in offering post-stroke neuroprotection. The calcium-activated potassium channel (K_Ca_3.1) that is expressed by microglia, cerebral vessel endothelial cells, and infiltrating monocytes in injured CNS has been reported to play an exacerbatory role in neuroinflammation [113]. Inhibitors of this channel cause a significant decrease in microglia production of nitric oxide and cyclooxygenase-2 (COX-2) [114,115].

Inhibition of Excitotoxicity: To minimize excitotoxicity, researchers targeted Ca^2+^ transport in the CNS (the neurons and blood vessels) by preventing calcium influx into neurons or by reducing extracellular Ca^2+^ availability using voltage-gated Ca^2+^ channel blockers (CCBs) and chelates. CCBs decreased ischemic insult in animal models of brain injury. The ability of these blockers to reduce stroke risk by 13.5% in comparison to diuretics and β-blockers in another study underscores their potential as preventive stroke therapy [116]. In addition, the highly lipophilic Ca^2+^ antagonist nimodipine was shown to enhance acute reperfusion in patients with acute ischemic stroke [117]. Furthermore, a Ca^2+^ chelate, DP-b99, proved safe and efficacious in phase I/II clinical trials when administered to stroke patients by significantly improving their clinical symptoms within 12 h of onset [118]. Despite the poor safety and efficacy profile of some sodium channel blockers in clinical stroke trials, mexiletine, which is also a Na^+^ channel blocker, proved effective in grey and white matter IS in animals, though further evaluation is required to confirm its role [119].

Prevention of Oxidative stress: This is a strong target for prospective IS therapies. Increasing evidence suggests that oxidative stress and apoptosis are closely linked phenomena in the pathophysiology of IS [120]. Progress has been made in the trial of free-radical-targeted agents as potential neuroprotective agents in stroke. Antioxidants with the ability to chelate iron or scavenge/trap free radicals have been examined in experimental IS models and clinically evaluated as neuroprotective agents [121]. Ebselen, a seleno-organic antioxidant with a glutathione-like mode of action, protects cellular structures from oxidative damage by scavenging reactive radicals and reacting with peroxynitrites, hydroperoxides, and thiols. The preclinical neuroprotective activity of ebselen has been replicated in numerous clinical trials including those of IS [121,122,123]. Although ischemic stroke patients treated with ebselen had a slightly better outcome, the observed difference in some studies was not statistically significant [124]. Alternatively, edaravone performed well in clinical trials [125]. Similarly, NXY-059, a spin trap agent, was neuroprotective in the rabbit model of embolic IS when combined with alteplase [126]. The neuroprotective effect of NXY-059 was not replicated in a phase III clinical trial NCT00119626 conducted by Lees et al., but there was a significant reduction in the occurrence of disability at 90 days [127]. Based on the findings from a larger study by the same researchers, NCT00061022, it was concluded that NXY-059 is devoid of therapeutic benefits in acute IS [128]. Deferoxamine, an iron chelate, that acts by reducing iron availability for hydroxyl radical production, has been shown to reduce oxidative stress and glutamate excitotoxicity-induced neurotoxicity [129,130]. Deferoxamine showed neuroprotective potentials in an unpublished pilot study in ischemic stroke patients [131].

Apoptosis Inhibition: Following IS, neurons in the ischemic penumbra are typically lost to apoptosis via intrinsic and extrinsic mechanisms [132]. Cdp-choline, a key intermediary in the biosynthesis of the important component of the neural cell membrane phosphatidylcholine, has shown antiapoptotic effects in cerebral ischemia. An exogenous form of CDP-choline is citicoline, whose mechanism of action includes inhibition of some phospholipases, increasing the availability of some catecholamines in the brain, and stimulating the synthesis of neuronal membrane phospholipids. In a meta-analysis conducted by pooling the individual patients’ data from several clinical trials in acute stroke, citicoline showed a significant increase in the odds of recovery at 3 months compared with placebo [133]. The meta-analysis findings were confirmed in the phase III clinical trial in which, in addition to being safe, citicoline caused remarkable improvement in functional, global, and neurological outcomes in IS patients [134]. Additionally, there are emerging therapies that target other forms of apoptosis such as necroptosis. Necroptosis inhibitors, such as Nec-1 and GSK’872, have shown promising results in preclinical studies by reducing infarct volume and improving neurological deficits [135].

Mitochondria stabilization: Neuroprotection via apoptosis inhibition has been achieved by inhibitors of mitochondrial permeability transition pores (MPTP). Mitochondria are considered the main link between cellular stress signals and the execution of programmed neuronal death [136,137]. Translocation of cytochrome c (Cyt c) from mitochondria to the cytoplasm is a key step in the initiation and/or amplification of apoptosis. Calcium-induced Cyt c release, as occurs in neurons during stroke and ischemia, involves rupture of the mitochondrial outer membrane and can be blocked by inhibitors of MPTP, thus preventing astrocyte activation [138]. MPTP blockers, such as cyclosporin A (CsA) and bongkrekic acid, have shown neuroprotective effects in animal models of ischemia and IS clinical trials [139,140,141,142].

Moreover, the dynamic morphology of mitochondria is retained through two opposite processes—fission and fusion. While the fission process involves the constriction and cleavage of mitochondria, the elongation of mitochondria via the joining and tethering of the mitochondria in close proximity constitutes the fusion process [143,144,145,146]. Accumulating evidence indicates that the maintenance of the mitochondrial function is crucial for neuron survival and neurological improvement. Therefore, targeting mitochondria is one of the promising neuroprotective strategies for stroke treatment [147]. Dynamin-related protein 1 (Drp1) is a mitochondrial-binding GTPase that mediates mitochondrial fission [148]. Maintaining mitochondrial dynamics has emerged as a crucial process in the regulation of cell survival and death, particularly as the fission process precedes neuronal death after cerebral ischemia [145,146]. Expectedly, Drp1 inhibitors such as mdivi-1, mdivi-A, and mdivi-B were shown to reduce infarct volume in a focal cerebral Ischemia model [147,148,149] (Figure 3).

## 5. Future Perspectives: Drug Repurposing and Re-Designing

In the face of limited time and financial resources coupled with the urgent need for newer IS therapies, drug repurposing presents an opportunity for the rapid development of safe and efficacious drugs for acute IS treatment and the optimal management of post-stroke sequelae. This approach involves either utilizing drugs that are approved for treating certain disease conditions for another disease/disorder with similar pathophysiology or using safe drugs that failed to improve clinical outcomes in one condition and trying them in a different but related disease condition. Many of the currently available neuroprotective agents were discovered via drug repositioning. Drugs like deferoxamine, citicoline, cyclosporin A, and nimodipine, which have been established for the management of iron poisoning, Parkinsonism, post-transplant immunosuppressant, and ischemic heart disease, respectively, have now found use as neuroprotectants in IS patients. A wide range of drug classes are currently being considered for the treatment and management of IS. Antidiabetics, immunomodulatory, calcium channel blockers (CCB), potassium channel blockers (PCB), opioid antagonists, antibiotics, and xanthine oxidase inhibitors constitute some of the drug classes with member(s) that have been tried for IS management.

Metformin, glibenclamide, glyburide, and saxagliptin are antidiabetic agents with promising use as post-stroke neuroprotectives. These drugs generally confer neuroprotection by regulating oxidative stress and modulating the AMPK/mTOR signaling pathway [150,151,152,153]. In addition, saxagliptin and other dipeptidyl-peptidase-4 inhibitors prevent inflammation by activating CXCR4/stromal-derived factor-1α pathway [153]. Interestingly, metformin, because of its biguanide moiety, drives microglia to the anti-inflammatory M2 phenotype, resulting in immune modulation [154]. Like the peripheral nerves, central nerve endings contain calcium channels that regulate Ca^2+^ transport. CCBs prevent excitotoxicity, and the neuroprotective effects of amlodipine and nimodipine have been reported in acute IS patients [117,155]. The neuroprotective effect of ziconotide, a conopeptide approved by the FDA for the management of chronic intractable pain, has been shown in several animal stroke models. Ziconotide inhibits N-type calcium channels, suppressing neural activity, and has since demonstrated neuroprotective potential in various animal stroke models [156,157,158,159]. Senicapoc, a K_Ca_3.1 channel inhibitor initially designed for treating vaso-occlusive crisis in sickle cell anemia, now shows potential for alleviating post-stroke inflammation due to its ability to cross the blood–brain barrier [115,160,161]. In addition to preventing post-stroke infections, reports suggest that antibiotics like minocycline and azithromycin reduce infarct volume, decrease the brain levels of matrix metalloproteinase-9 (MMP-9), and drive microglia to an anti-inflammatory phenotype [162,163]. Minocycline in particular may also act by inhibiting oxidative stress, apoptosis, and glutamate-induced excitotoxicity [163,164]. Naloxone, naltrexone, β-funaltrexamine, and allopurinol are other drugs with reported neuroprotective potential in IS [165,166,167] (Table 2).

The table highlights certain features of some drugs that have found use in the management of post-stroke sequelae. These drugs act by targeting at least one of the main pathogenetic pathways in ischemic stroke.

Despite the preclinical success recorded for some IS pathophysiology-targeting drugs, a handful of these drugs could not replicate their remarkable preclinical performance in human stroke volunteers. For instance, lubeluzole, a sodium-channel blocker and an NMDA receptor antagonist, was shown to reduce mortality in stroke in initial clinical trials, but successive trials failed to reproduce similar outcomes. Moreover, the required dose for neuroprotection was too high and is associated with a significant possibility of heart failure [168,169]. Similarly, sipatrigine, a neuronal Na^+^- and Ca^2+^-channel blocker, failed in a phase II clinical trial in stroke patients despite remarkable results in preclinical stroke models. Additionally, amiodarone was shown to aggravate brain injury due to defective transportation and accumulation of Na^+^ ions in the brain after stroke [170]. As it is the main inhibitory neurotransmitter in the brain, it was thought that elevating the level of gamma amino butyric acid (GABA) would counteract glutamate-induced excitotoxicity that occurs after an IS attack, thus preventing neuronal death. However, the use of the GABA agonist clomethiazole failed to reduce the glutamate-induced toxicity in stroke patients in clinical trials [171]. Drug redesign offers a plausible approach to circumvent challenges like this.

Redesigning a drug often involves purposeful alteration of its structure to obtain desired physicochemical properties. Drugs can be redesigned to modify their solubility, receptor-site specificity, enzyme affinity, and therapeutic target, and enhance safety/tolerability. Nimodipine is a dihydropyridine L-type calcium channel blocker that has high affinity for cerebral arteries where it acts to dilate them and cause recanalization. The difference between nimodipine and other CCBs lies in the chemical structure of the former having more alkyl moiety attached to the pyridine ring to make it highly lipophilic and able to cross the BBB. Similarly, the structure of ziconotide can be manipulated to enhance its lipophilicity, thus reducing the amount needed to achieve desired post-stroke neuroprotection in IS patients, and invariably reducing its side effects. Similarly, the peripheral cardiac effect associated with high doses of lubeluzole can be mitigated through structural modification. Bulky groups can be added to lubeluzole’s aromatic rings to enhance its lipophilicity. This modification can likely improve its blood–brain barrier penetration, reducing the required dosage for neuroprotection in IS.

## 6. Conclusions

Factors such as the complexity of ischemic injury, heterogeneity of patient population, limited therapeutic window for effective neuroprotection, and unforeseen side effects make translating experimental findings to clinical therapies a challenge. Considering the multifactorial nature of ischemic injury, exploring combination therapies that target multiple pathways simultaneously may enhance the likelihood of success in clinical translation. Moreover, refining the preclinical stroke models to better mimic the complexity of human stroke may increase the predictive value of preclinical studies. Nonetheless, a comprehensive understanding of ischemic stroke pathophysiology has been and will continue to inspire the development of novel drugs for its treatment. In the era of in silico drug design offering high-throughput screening of innumerable drugs at a go, it is much easier now to discover IS drugs from existing drugs and from de novo synthesis. The hope to increase the pool of effective therapy for the prevention and management of IS is thus not lost.

CNS delivery of drugs, especially those with poor lipophilicity, is often a challenge. Thankfully, emerging technologies are presenting novel and effective ways to circumvent them. Improving the delivery of natural antioxidants to the CNS for ischemic stroke treatment involves employing advanced modalities. Nanoparticle-based systems, such as liposomes and polymeric nanoparticles, enable targeted and sustained release, overcoming the blood–brain barrier (BBB). Ligand-targeted nanocarriers and intranasal administration provide non-invasive alternatives to enhance antioxidant access to the CNS. Prodrug strategies and innovative formulations like nanosuspensions and hydrogels improve solubility and stability. Focused ultrasound with microbubbles temporarily disrupts the BBB. Additionally, microneedles and combination therapies offer promising avenues for optimizing the multimodal mechanisms of action for effective antioxidant delivery in ischemic stroke.

## Figures and Tables

**Figure 1 biomolecules-14-00305-f001:**
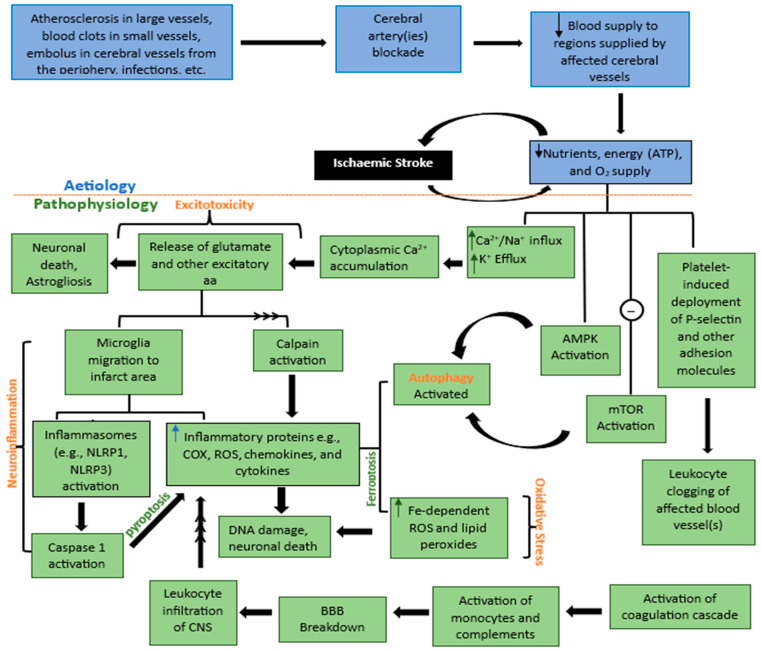
Pathogenesis of Ischaemic Stroke. In the event of cerebrovascular obstructions, blood supply is halted leading to a fall in energy (ATP), nutrient, and oxygen supply to the affected brain region(s). Consequently, a cascade of events occurs—an imbalance in calcium, sodium, and potassium ions, deployment of adhesion molecules, activation of AMPK, and inhibition of mTOR activation. These diverse events result in calcium accumulation and subsequently excitotoxicity, neuroinflammation, oxidative stress, apoptosis, and autophagy, with each occurring separately and supporting the other to cause cell death. Key: aa = Amino acids, Fe = Iron, ATP = Adenosine triphosphate, Ca^2+^ = Calcium ions, Na^+^ = Sodium ions, potassium ion = K^+^, COX = Cyclooxygenase enzyme, ROS = Reactive oxygen species, AMPK = Adenosine monophosphate-activated protein kinase, mTOR = mammalian target of rapamycin, arrow pointing up = rise/increase/upregulation, arrow pointing down = drop/decrease/decline/downregulation, arrows with multiple heads = series of events.

**Figure 2 biomolecules-14-00305-f002:**
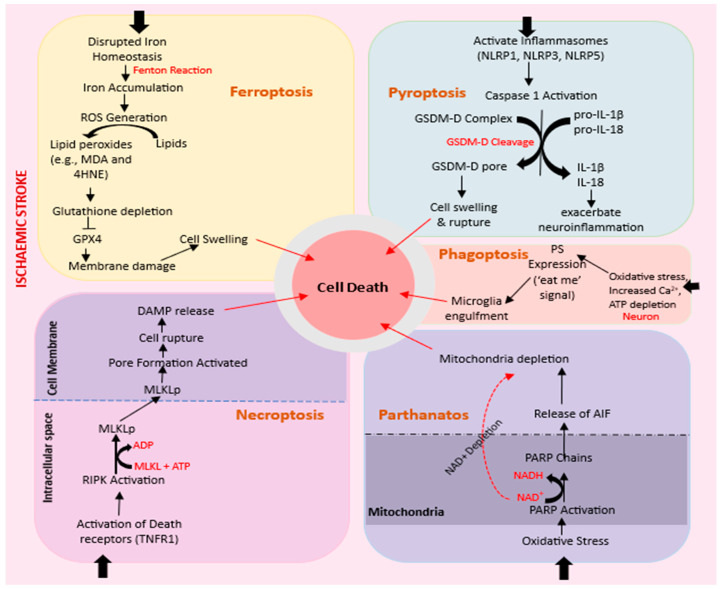
Different Forms of Cell Death Associated with Ischemic Stroke.

**Figure 3 biomolecules-14-00305-f003:**
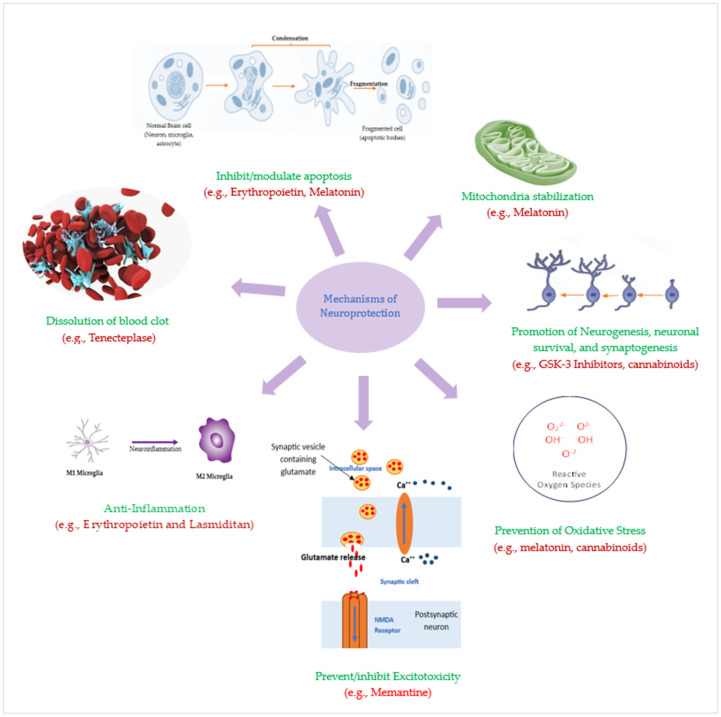
Mechanism of Neuroprotection by Potential Neuroprotective Therapies.

**Table 1 biomolecules-14-00305-t001:** Stem cell therapy for ischemic stroke in clinical trials.

S/No	Identifier	Stem Cell Type	Study Centre	Study Aim	Clinical Trial Phase	Study Status
1	NCT00875654	Autologous mesenchymal stem cells	France, Europe	Feasibility and tolerance	Phase I	Completed
2	NCT05008588	Umbilical cord mesenchymal stem cells (whole cell and conditioned medium)	Indonesia, Asia	Safety and Efficacy	Phase I/II	Ongoing
3	NCT02117635	Allogeneic human neural stem cell	United Kingdom, Europe	Efficacy	Phase II	Completed
4	NCT01501773	Autologous bone marrow stem cell	India, Asia	Safety, feasibility, and efficacy	Phase II	Completed
5	NCT04811651	Umbilical cord-derived mesenchymal stem cells	China, Asia	Safety and efficacy	Phase II	Recruiting
6	NCT01716481	Autologous mesenchymal stem cells	South Korea, Asia	Neuroprotection	Phase III	Completed
7	NCT04631406	Neural stem cell	California, North America	Safety and tolerability profile	Phase I/II	Recruiting
8	NCT03356821	Stromal cells (intranasal)	Netherlands, Europe.	Safety and feasibility	Phase I/II	Completed
9	NCT05292625	Umbilical cord-derived MSC (intrathecal and intravenous)	Vietnam, Asia	Safety and efficacy	Phase I/II	Recruiting
11	NCT06138210	Induced pluripotent stem cell	China, Asia	Safety and preliminary efficacy	Phase I	Starts 2024
12	NCT00859014	Autologous mononuclear bone marrow cells (intravenous)	Houston, North America	Safety and tolerability	Phase I	Completed
13	NCT02178657	Autologous mononuclear bone marrow cells (intra-arterial)	Spain, Europe	Safety and neuroprotection	Phase II	Ongoing
14	NCT00950521	CD34+ stem cell (intracerebral implantation)	Taiwan, Asia	Efficacy	Phase II	Completed
15	NCT00535197	Autologous CD34+ subset bone marrow stem cell (intra-arterial infusion)	London, Europe	Safety and tolerability	Phase I/II	Completed

**Table 2 biomolecules-14-00305-t002:** Some repurposed drugs for IS management and their pathophysiological targets.

	Drugs	Drug Class	Approved Indication(s)	Pathophysiology Target(s)
Wang et al., 2007 [155]	Amlodipine	Calcium channel blocker (CCB)	Hypertension, Angina	Excitotoxicity
Smith et al., 2019 [162]	Azithromycin	Macrolide antibiotic	Sinusitis, conjunctivitis, community-acquired pneumonia	Neuroinflammation
Cho & Kim, 2009 [134]	Citicoline	Neurotropic	Parkinsonism, head injury	Apoptosis
Forsse et al., 2019 [139]	Cyclosporin A	Immunomodulator	Post-transplant immunosuppression	Mitochondrial dysfunction secondary to oxidative stress and excitotoxicity
Selim et al., 2009 [131]	Deferoxamine	Chelate	Iron poisoning	Excitotoxicity, oxidative stress, ferroptosis
King et al., 2018 [152]	Glyburide	Sulfonylureas antidiabetic	Diabetes mellitus	Neuroinflammation and oxidative stress
Zhao et al., 2019 [151]	Metformin	Biguanide antidiabetic	Diabetes mellitus, polycystic ovary syndrome	Neuroinflammation and oxidative stress
Kikuchi et al., 2012 [164]	Minocycline	Tetracycline antibiotic	Inflammatory acne, gonococcal infections, urinary tract infection	Excitotoxicity, neuroinflammation, apoptosis, and oxidative stress
Anttila et al., 2018 [167]	Naloxone	Opioid receptor antagonist	Opioid overdose	Neuroinflammation
Staal et al., 2017 [161]	Senicapoc	Calcium-dependent potassium channel (K_Ca_3.1) blocker	Sickle cell anemia	Neuroinflammation
Twede et al., 2009 [156]	Ziconotide	Conopeptide, analgesic, N-type CCB	Chronic intractable pain	Excitotoxicity

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
