# Peer review of "Understanding the Pathophysiology of Ischemic Stroke: The Basis of Current Therapies and Opportunity for New Ones"

_biomolecules, 2024, doi:10.3390/biom14030305_

Round 1

Reviewer 1 Report

Comments and Suggestions for Authors

This manuscript comprehensively discusses the pathological mechanisms of ischemic stroke that lead to neuronal apoptosis and functional injury, and explores potential therapeutic agents, protocols and measures. This manuscript has certain promotion significance for understanding the pathological mechanism of ischemic stroke, but some contents need to be improved and supplemented.

1. The pathological mechanism diagram in figure 1 lacks other mechanisms discussed in this manuscript, such as Ferroptosis, Necroptosis, Parthanatos and Phagoptosis. Please add relevant contents or draw new neuronal injury mechanism diagrams.

2.In section of the Neuroprotective Agents for Ischaemic Stroke, potential therapeutic agents and mechanisms with neuroprotection need to be described in detail or shown in a diagrams.

3.A large number of studies have been conducted on reducing nerve injury and promoting neuroprotection after ischemic stroke in experimental animals. However, few studies can be translated into the clinic therapies. Please discuss the reasons in the conclusion section.

4.The disturbances in biological processes such as mitochondrial dynamics, and mitochondrial quality control play an important role in the pathological process of ischemic stroke. In the pathological mechanisms and potential treatments section, please discuss the pathology and therapeutic mechanisms of ischemic stroke in conjunction with mitochondrial quality control and mitochondrial dysfunction, such as mitochondrial dynamics, mitochondrial transplantation, etc.

Author Response

Thank you very much for taking the time to review this manuscript. We are sincerely grateful for the insightful feedback. Please find the detailed responses below and the corresponding revisions/corrections in track changes in the re-submitted files

Comment 1. The pathological mechanism diagram in figure 1 lacks other mechanisms discussed in this manuscript, such as Ferroptosis, Necroptosis, Parthanatos and Phagoptosis. Please add relevant contents or draw new neuronal injury mechanism diagrams.

Response 1: Thank you for pointing this out. We agree with this comment, and we’ve included a short paragraph to capture this (lines 203-208). We have also included a new figure to summarize these death pathways with a comprehensive legend attached to it (see Figure 2, page 7)

Comment 2: In section of the Neuroprotective Agents for Ischaemic Stroke, potential therapeutic agents and mechanisms with neuroprotection need to be described in detail or shown in a diagrams.

Response 2: Thanks for this suggestion. However, we believe this has been explicitly captured in section 4.0. A new figure (Figure 3) summarizes the mechanisms.

Comment 3. A large number of studies have been conducted on reducing nerve injury and promoting neuroprotection after ischemic stroke in experimental animals. However, few studies can be translated into the clinical therapies. Please discuss the reasons in the conclusion section.

Response 3: We couldn’t agree more with this submission and as such, we have briefly highlighted these reasons in the opening paragraph of the conclusion section (see lines 663 -669).

4.The disturbances in biological processes such as mitochondrial dynamics, and mitochondrial quality control play an important role in the pathological process of ischemic stroke. In the pathological mechanisms and potential treatments section, please discuss the pathology and therapeutic mechanisms of ischemic stroke in conjunction with mitochondrial quality control and mitochondrial dysfunction, such as mitochondrial dynamics, mitochondrial transplantation, etc.

Response 4: Brilliant observation! we have included the involvement of mitochondria in the pathology of ischaemic stroke on page 2 of the revised submission (lines 123-133).  Mitochondria as a therapeutic target has been captured on page 13 (lines 573-584)

Reviewer 2 Report

Comments and Suggestions for Authors

The manuscript addresses a hot topic and is generally well written. However, a series of issues need revision according to the following suggestions:

In the abstract:

- the fragment “a few or most of their…” is too encompassing and I think it could be deleted

- “minimal availability of effective therapy” is not correct. Recanalization (mainly thrombectomy) can have amazing results if feasible and accessible.

Page 2, line 59 – the occlusion of a cerebral artery does not occur by the fatty deposits of the atherosclerotic plaque, but to a thrombus formed on top of this plaque due to a series of events that lead to thrombocytes adhering and aggregating on the plaque’s surface

Pg 2 line 61 – other manifestations may occur as well; I think that it should be highlighted that depending on the territory supplied by the occluded artery, a suggestive combination of motor, sensory, or speech disturbances, as well as disturbed coordination may occur.

Pg 2, line 63 – it should be explained what ischemic core and penumbra means – ref:

DOI: 10.31083/j.jin2003078

Pg 4 – autophagy is an energy-consuming process that cannot occur in the ischemic core, but can contribute in the penumbra o delayed cell loss, or occur after reperfusion

Pg 4, lines 144-146 – the 2 sentences convey the same idea; one should be removed.

Pg 5 – the therapeutic attempts to prevent necroptosis should be covered in the Treatment section

Pg 6 – there is no discussion on mechanical thrombectomy for large vessel occlusion, though ref 65 and 66 describe the add-on benefits of the antithrombotic agents used in conjunction with thrombectomy

Pg 6 – antiplatelet agents: I think that a brief overview of the mechanism of action of these antiplatelets would be useful, as well as a warning on the risk of increasing the rate of cerebral hemorrhages if used for more than 3 months in combination (findings of the MATCH and CHARISMA studies)

Pg 7, lines 259-260 – defibrinogen agent – replaced with defibrinogenating agent

Pg 7 – section on stem cell therapy – some comments on the delivery methods of each of the categories of stem cells, potential side effects would be useful (suggested ref: DOI: 10.1002/sctm.19-0076, doi: 10.3390/bioengineering9110717, DOI: 10.1016/j.nano.2020.102149), as would the possibility of delivering stem cell-derived extracellular vesicles, with the possibility to engineer miRNAs contained in these vesicles.

Pg 8, lines 303-306 have already been covered, should be removed.

Lines 321-323 – CCBs reducing the risk of stroke proves their usefulness in stroke prevention, not treatment

Pg 9, line 340 – although ischemic stroke patients treated with ebselen had a slightly better outcome, the difference failed to reach statistical difference (DOI: 10.1161/01.str.29.1.12). Alternatively, Edaravone performed well in clinical trials ( DOI: 10.1016/j.jns.2011.09.006)

Pg 11 line 421 – despite remarkable results

Pg 11 – I believe that a brief discussion of the modalities to improve CNS delivery of natural antioxidants with multimodal mechanism of action using nanomedicine technologies would benefit the manuscript

NOTE; the quoting mode of the references do not comply with the requirements of the journal and should be modified. In addition, using subtitles and numbered paragraphs could help navigate through the manuscript.

Author Response

Thank you very much for taking the time to review this manuscript. We are deeply grateful for the invaluable feedback. Please find the detailed responses below and the corresponding revisions/corrections in track changes in the re-submitted files

Comments: In the abstract: 

- the fragment “a few or most of their…” is too encompassing and I think it could be deleted 

Response: Noted and removed.

- “minimal availability of effective therapy” is not correct. Recanalization (mainly thrombectomy) can have amazing results if feasible and accessible. 

Response: True. We have changed the statement to read thus “…..minimal availability of pharmacological therapies…”

Page 2, line 59 – the occlusion of a cerebral artery does not occur by the fatty deposits of the atherosclerotic plaque, but to a thrombus formed on top of this plaque due to a series of events that lead to thrombocytes adhering and aggregating on the plaque’s surface 

Response: True. Noted and corrected.

Pg 2 line 61 – other manifestations may occur as well; I think that it should be highlighted that depending on the territory supplied by the occluded artery, a suggestive combination of motor, sensory, or speech disturbances, as well as disturbed coordination may occur. 

Response: we couldn’t agree more and have added this bit of info (lines 62 - 63)

Pg 2, line 63 – it should be explained what ischemic core and penumbra means – ref: American Journal of Neuroradiology January 2006, 27 (1) 20-25; 

Response: This has been corrected in section 2.0, lines 60 to 64

Pg 2, lines 64-69 – the authors refer too many times to excitotoxicity, oxidative stress, neuroinflammation – it is redundant. The fragment should be reorganized and .use subtitles and headings. 

Response: Thanks for this observation. We have separated the three mechanisms and have discussed them as appropriate.

Pg 2, line 73 – the brain requires constant supply of oxygen and glucose, not merely circulation or blood flow. The reduced ATP production interferes with the normal functioning of the ion pumps, leading to membrane depolarization and release of glutamate, which, by acting on NMDA and AMPA receptors, can propagate the molecular disturbances to neighboring and distant sites. 

Response: Duly noted and has been captured in the updated discussions for excitotoxicity

Pg 2, lines 86-88 – I don’t think that in this section the authors should refer to therapies (miRNAs), since they have a large part dealing with therapeutic strategies. 

Response: This has been moved to section 4.0 under “mitochondria stabilization” (lines 563 to 586)

Pg 3-4. Apoptosis can use the intrinsic or extrinsic pathway – ref: DOI: 10.31083/j.jin2003078 

Response: Sure. We already captured this in the discussion of apoptosis

Pg 4, lines 144-146 – the 2 sentences convey the same idea; one should be removed. 

Response: That was an oversight. Thank you for pointing it out. The latter has been removed

Pg 5 – the therapeutic attempts to prevent necroptosis should be covered in the Treatment section 

Response: This was initially included under necroptosis but has been moved to section 4.0, lines 559-562

Pg 6 – there is no discussion on mechanical thrombectomy for large vessel occlusion, though ref 65 and 66 describe the add-on benefits of the antithrombotic agents used in conjunction with thrombectomy 

Response: Mechanical thrombectomy has now been properly captured on page 9 (lines 357 -367)

Pg 6 – antiplatelet agents: I think that a brief overview of the mechanism of action of these antiplatelets would be useful, as well as a warning on the risk of increasing the rate of cerebral hemorrhages if used for more than 3 months in combination (findings of the MATCH and CHARISMA studies) 

Response: Sure. An overview of the moa of the mentioned antiplatelets has been included in lines 389– 397. Findings from the MATCH and CHARISMA trials have also been briefly highlighted in lines 403 -415

Pg 7, lines 259-260 – defibrinogen agent – replaced with defibrinogenating agent 

Response: Noted and corrected.

Pg 7 – section on stem cell therapy – some comments on the delivery methods of each of the categories of stem cells, potential side effects would be useful (suggested ref: DOI: 10.1002/sctm.19-0076, doi: 10.3390/bioengineering9110717, DOI: 10.1016/j.nano.2020.102149), as would the possibility of delivering stem cell-derived extracellular vesicles, with the possibility to engineer miRNAs contained in these vesicles. 

Response: These have been covered in a new section (3.3.1)

Pg 8, lines 303-306 have already been covered, should be removed. 

Response: Removed!

Lines 321-323 – CCBs reducing the risk of stroke proves their usefulness in stroke prevention, not treatment 

Response: Sure. The statement has been adjusted to capture this.

Pg 9, line 340 – although ischemic stroke patients treated with ebselen had a slightly better outcome, the difference failed to reach statistical difference (DOI: 10.1161/01.str.29.1.12). Alternatively, Edaravone performed well in clinical trials ( DOI: 10.1016/j.jns.2011.09.006) 

Response: Interesting! We found this observation quite insightful and have added it to our discussion of oxidative-stress-targeted therapies

Pg 11 line 421 – despite remarkable results 

Response: *Smiles* This has been corrected. Thank you!

Pg 11 – I believe that a brief discussion of the modalities to improve CNS delivery of natural antioxidants with multimodal mechanism of action using nanomedicine technologies would benefit the manuscript 

Response: We couldn’t agree more and have captured this in the closing paragraph of the conclusion section.

NOTE; the quoting mode of the references do not comply with the requirements of the journal and should be modified. In addition, using subtitles and numbered paragraphs could help navigate through the manuscript. 

Response: Brilliant suggestion. We’ve corrected the referencing style and have implemented your suggestion concerning paragraph numbering in the reviewed manuscript. Thank you.